# Evaluation of the Loss of Strength, Resistance, and Elasticity in the Different Types of Intraoral Orthodontic Elastics (IOE): A Systematic Review of the Literature of In Vitro Studies

**DOI:** 10.3390/jpm13101495

**Published:** 2023-10-14

**Authors:** Sabina Saccomanno, Vincenzo Quinzi, Licia Coceani Paskay, Livia Caccone, Lucrezia Rasicci, Eda Fani, Daniela Di Giandomenico, Giuseppe Marzo

**Affiliations:** 1Department of Health, Life and Environmental Science, University of L‘Aquila, 67100 L’Aquila, Italy; vincenzo.quinzi@univaq.it (V.Q.); livia991@yahoo.it (L.C.); rasiccilucrezia@gmail.com (L.R.); daniela.digiandomenico@gmail.com (D.D.G.); giuseppe.marzo@univaq.it (G.M.); 2Academy of Orofacial Myofunctional Therapy (AOMT), Pacific Palisades, CA 90272, USA; lcpaskay@gmail.com

**Keywords:** intraoral orthodontic elastics, loss of strength, latex and synthetic elastics, force, in vitro

## Abstract

Background: Intraoral orthodontic elastics (IOE), typically referred to as rubber bands, are important tools for correcting malocclusion, and they are classified into latex and synthetic (elastomeric-based) elastics. They have different strengths and sizes, depending on their intended use, that provide clinicians with the ability to correct both anteroposterior and vertical discrepancies. Clinical use, together with saliva, alters the physical characteristics of both latex and synthetic elastics, causing declines in strength over time. Aim: The aim of the study was to assess, through a systematic review of in vitro studies, the properties of intraoral elastics. The primary goal was to evaluate how IOEs behave in terms of tension strength and duration. The secondary goal was to investigate the force loss during the first hours of wear. The tertiary goal was to assess how these forces decayed. Materials and methods: The following electronic databases were searched from December 2020 to April 2021: Medline Full Text, PubMed, Cochrane Clinical Trials Register, Science Direct, and Literature Review. Out of 8505 initial articles, 10 were selected for the systematic review. Results: The force-degradation property was found in all types of IOEs. The loss of strength was directly proportional to time, with the highest value during the first 3 h after extension, regardless of the elastic band size and manufacturer. The forces generated by the latex bands were higher than in those of the elastomeric-based elastics, but they did not consistently correspond to the loads specified by the manufacturers. The retention forces in the latex IOEs were significantly higher than those in the nonlatex bands, suggesting that elastomeric-based bands need to be changed more frequently and at regular intervals throughout a 24 h period. Conclusion: This systematic review indicates that intraoral orthodontic elastics have the greatest loss of force during the first 3 h, that latex rubber bands have the highest strength during the first hour, that the forces generated are not always consistent with the manufacturer’s specifications, and that nonlatex (elastomeric-based) IOEs need to be changed frequently and regularly during a 24-h cycle.

## 1. Introduction

Orthodontic treatments allow for the treatment of malocclusions to create an aesthetic and occlusal harmony [1]. The types of orthodontic treatments include both fixed and mobile (aligners) treatments. The fixed types use brackets, metal alloy bands on molars, and preformed wires to move teeth into a desired position [2]. The wires are held against the brackets and molar bands by way of elastics or metal ligatures, but elastic bands are used to pull the brackets and bands in the desired position. The elastic bands are made of silicone or latex (elastomeric-based) materials.

Orthodontic elastic bands are a key component of orthodontics as they allow for the movement of dental arches to make them fit together in optimal occlusion. In the 1970s, the orthodontist Charles H. Tweed was among the first to use elastics in a specific way in fixed orthodontic treatments, namely, to close the post-extraction spaces of premolars [2].

According to the standard ISO 21606:2022, OIE are elastomeric auxiliaries, which are defined as “devices with elastomeric properties for transmission of forces” used as orthodontic devices, and they include:(1)Orthodontic elastics (usually referred to as “rubber bands”), which are single rings used intra- and extraorally to apply specific forces;(2)Orthodontic elastomeric chains in which rubber bands are linked together, as opposed to being separate;(3)Orthodontic threads, which are hollow threads used when the distances between brackets are greater than the size of an orthodontic elastic;(4)Orthodontic elastomeric ligatures, which replace the metal ligatures used to connect a wire to a bracket or a molar band;(5)Orthodontic elastomeric separators, which are positioned between teeth to separate them [3].

For the sake of this study, we took into consideration only intraoral orthodontic elastics (IOEs). Also, for the sake of this study, we refer to the following:(1)Initial extension force (F0) as the force applied to an orthodontic elastic by stretching it to three times its length after an initial stretch to four times its length;(2)Twenty-four-hour residual force (F24) as the percentage of force exerted by an IOE when stretched to three times its test length at 24 h after an initial stretch to four times its test length;(3)Ultimate extension (A) as a percentage of the extension at which an IOE breaks compared to its test length [3].

The types of rubber bands chosen for therapy depend on the tooth and jaw movements that will be needed to achieve optimal alignment, and they are distinguished by their sizes, strengths (light, medium, and complex), and elastic lengths.

The therapeutic time is dictated by a patient’s compliance and by the type of occlusal problem to be solved. Many studies have indicated that the correct time of use for an IOE is 22 h per day, that they need to be changed at least every 24 h, and that the maximum loss of strength occurs within the first 4–5 h after application. Some patients will need to wear elastics from the start of their treatment while others will only need to wear them sometimes during treatment [4].

One of the disadvantages of elastics is that strength levels decrease over time. This property is called “strength decay”, and for optimal orthodontic tooth movement, this strength decay must be within acceptable limits. According to most manufacturers, the maximum force of an elastic is expressed by stretching the device to three times its diameter [4,5].

Prolonged use of an elastic can lead to its wearing, force degradation (loss of force over time), and eventual breakage, and it is preferable to change them at least once a day; otherwise, the elasticity can wear off, resulting in diminished strength until they snap [6,7]. However, there is no consensus on the frequency for changing elastic bands.

Even the presence of solutions within the oral cavity, such as mouthwashes, toothpastes, or sugary foods and drinks, can counteract the work performed by the rubber bands used.

The result of the orthodontic treatment is closely related to the use of the IOE; therefore, another limitation to the effectiveness of elastic bands is related to patients’ noncompliance or incorrect use, as these bands can, in some cases, be difficult to position and create discomfort for the patient right after their placement.

Intraoral elastics are normally manufactured with natural latex, a material without impurities and with precise tolerances in diameters. “Latex-free” intraoral elastics (elastomeric-based) are also available, specifically for patients allergic to latex, and usually, but not always, made of silicone.

Their main feature, which determines their effectiveness, is represented by elasticity, defined as the ability to return to the original shape after undergoing substantial deformation. Elasticity is determined by the geometric pattern and the type of molecular attraction existing in the rubber bands. However, there are limits, since a highly elastic material can cease to be so when the applied forces exceed certain values.

The aim of this study was to evaluate, through a systematic review of the literature of in vitro studies, the properties of intraoral elastics, and therefore the loss of strength, resistance, and elasticity in different types of elastics, and which ones maintain these properties the longest over time.

## 2. Materials and Methods

### 2.1. Guidelines

The PRISMA (Preferred Reporting Items for Systematic Reviews and Meta-Analysis) statement was adhered to as much as possible [8,9].

### 2.2. PICO Question

This review was designed following PICO guidelines to determine the properties of intraoral elastics.

Population: NA (they are all in vitro studies);

Intervention: analysis of intraoral elastics in vitro;

Comparison: intraoral elastics of different strengths, diameters, and materials;

Outcomes: The primary outcome was to evaluate how intraoral elastics behave in terms of tension strength and duration. The secondary outcome was to investigate the force degradation during the first hours of wear. The tertiary outcome was to assess how forces decayed.

### 2.3. Search Strategy

The protocol of this research was registered in the publicly accessible PROSPERO (CRD42021226680), the primary database for registering systematic review protocols.

A review of the bibliography on in vitro studies was conducted, with a search ranging from December 2000 to April 2021 through the following databases: Medline full text, PubMed, Cochrane Clinical Trials Register, Science Direct, and Literature Review.

The database search terms/keywords mentioned above were “force decay” and “force loss” and “orthodontic elastics”, “rubber band”, “elastics degradation”, “elastic force”, “dry test”, “Latex elastics”, “non latex elastics”, “in vitro study”, “elastomeric properties”, “elastomeric in dentistry”, “elastomeric orthodontics”, “elasticity”, “rubber”, “elastomeric chain”, and “force degradation”.

The language restriction was not enforced, and the study references were chosen manually.

### 2.4. Study Selection 1

A.The selection process for this study took place in two stages. In the first stage, studies were considered according to the following inclusion criteria (A): Nonsolution in vitro studies;B.In vitro solution studies;C.Manufacturers;D.Material (latex/nonlatex);E.Strength of the rubber bands expressed in ounces/grams;F.Elongation time;G.Deformation of the elastic;H.Analysis of the loss of strength over time (hours/days);I.Force expressed by the elastics before and after stretching;J.Evaluation in the different elastics: which ones are those that keep the resistance the longest over time:
a.Elongation to two or three times their diameter;b.Resistance to friction;c.Permanent deformation;d.Tensile strength;K.All languages;L.Year of publication.

Although two studies included referred to both in vitro and in vivo samples, we only considered the data referring to the in vitro portion.

Only studies that met the previous inclusion criteria progressed to the second phase, and were expressed in a survey according to the following exclusion criteria (B):A.Studies before 2000;B.Studies that reported a break in the elastic;C.All meta-analyses and systematic reviews, because many of the articles we selected were cited there.

### 2.5. Study Selection 2

A total of 8505 articles were identified from database search, and one more record from other sources was identified in December 2020 and included in this study. After removal of duplicates (5625), 2881 articles remained. By entering the filters specified in the inclusion criteria in the five search engines, 276 articles were excluded because the full article was not available.

Of the remaining 2605 articles, after reading the abstracts, 2354 were excluded due to incomplete or not comparable data. Therefore, 251 articles were read, of which 241 were excluded because, although they dealt with the behavior of IOE, it was not clear how their force decreased during the first hours of wear.

In the end, ten articles were included in quantitative synthesis for systematic literature review.

The flow chart of Figure 1 shows the process used, which was based on the PRISMA guidelines for systematic reviews.

See Table 1; Table 2 for the list of the articles selected and included in the systematic review.

### 2.6. Data Screening and Extraction

All the authors independently performed initial screening. At this stage, the title and abstract, if available, were investigated by all of the authors. The articles were selected if considered relevant by all the reviewers. A detailed full-text analysis was performed by three reviewers (S.S., L.R., and L.C.) who extracted data from each study. There was no dissent related to data screening and extraction.

### 2.7. Data Analysis

This systematic review was carried out by using original studies, comparative studies, research journal articles and, once analyzed, we collected data regarding five areas, as listed in Table 3.

We based our work on these five groups of data collected from the selected articles.

### 2.8. Quality Assessment

Three reviewers (S.S., L.R., and L.C.) independently evaluated the risk of bias. The risk of bias is summarized in Table 4. This evaluation was carried out following the Cochrane-recommended approach for assessing the risk of bias in randomized controlled clinical studies, including four quality parameters: sequence generation, consideration of incomplete outcome data, freedom of selective outcome reporting, and other sources of bias [8,9,12].

### 2.9. Outcome Measures 

The primary outcome was to evaluate how IOEs behave in terms of tension strength and duration. The secondary outcome was to investigate the force degradation during the first hours of wear. The tertiary outcome was to assess how forces decayed. See Table 5 for details on the types of elastics mentioned in the studies reviewed.

### 2.10. Levels of Evidence

Levels of evidence and grade of recommendation followed the Grading Recommendations of the GRADE Working Group [8].

Our 10 final studies, published between 2000 and 2021, were all level 2B since we excluded reviews and meta-analyses, which would have been level 1A.

## 3. Results

### 3.1. Assessment of Risk of Bias

The studies were graded as having high, uncertain, or low risk, based on selection bias, performance bias, detection bias, attrition bias and reporting bias (see Table 4). The quality of individual studies was evaluated based on the categorized ranking of the Oxford Centre for Evidence-Based Medicine 2011 Levels of Evidence.

### 3.2. Effects of Interventions: In Vitro Analysis

The primary outcome of this study was to evaluate how IOEs behave in terms of tension strength and duration. The results of this systematic review indicate that cyclical tests show that retention forces in latex rubber bands are significantly higher in than nonlatex bands, suggesting that nonlatex bands need to be changed more frequently at regular intervals throughout the 24 h.

Moreover, cyclical testing of IOE, both latex and nonlatex, resulted in significantly greater loss of strength. Nonlatex rubber bands were more affected than their latex counterparts. This could be due to increased chain slippage at the molecular level due to repeated stretching, due to the extension beyond the elastic limit of the product, or a combination of both. The loop of the bands also caused a greater decrease in force at the start of the test, but the rate of force decay was statistically similar to that of the bands tested after the first hour.

The secondary outcome was to investigate the force degradation during the first hours of wear. The results of this systematic review indicate that the force-degradation property was found in all types of IOE. The strength of degradation is directly proportional to the time with the highest value during the first 3 h after extension, regardless of size and manufacturers.

The tertiary outcome was to assess how forces decayed. It was noted that the forces generated by the IOE do not consistently correspond to the loads specified by the manufacturers.

## 4. Discussion

### 4.1. Overall Discussion

The decay of force expressed by the elastic bands as orthodontic devices is the key problem in clinical use as it makes it difficult for the operator to assess the real loss of force and, therefore, to recommend to the patient an accurate schedule of replacement of the elastics.

A total of 251 articles were initially considered for our study aimed at analyzing force decay in IOE.

Specifically, after applying all the exclusionary parameters, we considered a total of 10 studies as we focused on the decay of the strength of IOE only in vitro (see Table 2), showing that IOEs were unable to provide continuous force over time, confirming the clinical experience of orthodontists worldwide.

Regarding the primary outcome of this study, which was to evaluate how IOEs behave in terms of tension strength and duration, we found the following data.

Kanchana and Godfrey found that the standard dry force extension index could be precisely applied only to a small group of rubber bands used [10].

Yang et al. found that the strength loss of latex IOE was greater in vivo than in vitro dry or in artificial saliva: the larger the internal diameter, the smaller the strength value [14]. Also, the same article reported that the force degradation of latex elastic in vivo is much greater than that in both air and artificial saliva. In vivo, the force value of the orthodontic latex elastics decreased sharply in the first hour. The larger the inner diameter and smaller the setting force value were, the slower the force decay was.

Gangurde, Hazarey, and Vadgaonkar found a loss of strength in all types of rubber bands, equal to 20% on the first day and 5–10% on the second day [5].

According to Gioka et al., the reduction of strength of the rubber bands is a parameter dependent on the material used, because the size and strength of the band did not have a significant effect on the variation in the extent of the force decay [7].

Lopez, Vicente and Bravo [13] reported that when significant differences between wet and dry environments were found, the greater force loss occurred in wet conditions. However, some studies failed to detect significant differences in force loss between wet and dry media. Also, the same study, it was found that the only significant difference found between Lancer^®^ elastics, with and without latex in dry conditions, was that the force loss was greater for latex-free elastics, which clinically suggests that “old” batches of elastics, especially nonlatex ones, could be less efficient than “new” batches [11,13].

In previous reports where the majority of the studies were in vitro studies, the experimental conditions could be controlled accurately and the results were also reproducible. However, in the oral cavity, the characteristics of elastics are affected by many routine factors such as oral activities, oral liquid environment, different foods, and some other undefined factors [10]. There is also no consistency in the type of saliva used in the articles considered, as artificial saliva has a very specific composition [14], while naturally occurring saliva has many more variables, including variable pH, which may affect the functionality of the IOE [11].

Another difference between behavior in elastics is the environment they work in; for instance, Wang et al. stated that different environments have different effects on the properties of orthodontic latex elastics, especially because the oral environment has the potential to plasticize such polymers [16].

Hwang and Cha indicated that the latex bands all followed a similar pattern of force degradation, whereas the silicone bands showed a greater increase in force decay as the extension length increased [17].

Finally, Mansour concluded that the use of 1/4” diameter elastics is sufficient to cover the range of forces in orthodontic treatments [4].

Regarding the secondary outcome of this study, which was to investigate the force degradation during the first hours of wear, we found the following data.

Kanchana and Godfrey found an immediate reduction in strength after the initial stretch and then it gradually continued to decrease over a period of 3 days. Overall strength degradation was 29.9% in the first hour, 32.6% at 24 h, and 36.2% at the end of the 3 days [10].

According to Gioka et al., nonlatex elastics showed more strength decay over time than latex ones: the force applied by the elastic within the first 3–5 h underwent a significant decrease, while a decline of 20–25% was observed in the 24 h period [7]. This study has a similar conclusion to the article by Klabunde and Grünheid [18].

Wang et al. identified that the most significant force degradation occurred in the first half an hour, in both in vivo and in vitro studies, but the magnitudes of the loss of strength were different between conditions [16].

Alavi et al. found that the loss of strength after one hour was 4.75% and after 24 h it was 19–28%. Furthermore, various factors can affect the elastics inside the oral cavity; for example, the oral pH significantly affects the rate of force loss, as pH levels above neutral increase the force-decay rate [6].

Results by Lopez et al. [13] showed that samples of latex and nonlatex elastic bands were subjected to continuous stretching, measuring force at 5 s, 8 h, and 24 h in both dry and wet conditions. This was one of the only two studies comparing specific brands of elastic bands, but this one considered different sizes as well. Lancer^®^ nonlatex was the only type of elastics that did not show a significant decrease in its initial elastic characteristics at 5 s after activation, at 8 h in wet conditions. Nevertheless, both GAC^®^ latex and nonlatex and Lancer^®^ latex did show significantly less force than initially, in wet conditions only, at the 8 h and at the 24 h marks [13].

For this reason, Lancer^®^ nonlatex bands are the best option among the elastic bands evaluated in this study if they are worn for up to 8 h, because the other types of elastics evaluated did not even maintain their initial characteristics at this time point (8 h). Notwithstanding, an in vivo study would be necessary in order to confirm these results.

Lopez et al. found that the force loss was greater for GAC^®^ elastics than for Lancer^®^ elastics, in wet conditions in elastics both with and without latex, while Lancer^®^ latex elastics generated forces at 8 h and at 24 h that were significantly less than the initial ones [13].

The study by Fernandes et al. found that the force decay pattern showed a notable drop-off of forces during 0 to 3 h, a slight increase of the force values from 3 h to 6 h, and a progressive force reduction between 6 h to 24 h [19].

Regarding the tertiary outcome of this study, which was to assess how forces decayed, we found the following data.

According to Yang et al., after 12 h, force degradation was lower in many groups, whereby the greater the internal diameter of the elastic, the lower the resistance value. [14] In fact, after 12 h, the loss of strength increased in many groups of bands, which would explain why, in clinical application, it is useful to change the rubber bands at least every 12 h. From a clinical perspective, it seems to be important to discard old batches of elastics, especially the nonlatex ones as they tend to degrade in dry conditions.

Summary of connections to clinical utility:

The studies taken into consideration for this systematic review highlight some clinically useful information, such as that degradation over time of all types of elastics seems inevitable [13]. It would be ideal if the manufacturers would provide a “best by” date or preservation recommendations for elastics (such as light exposure, temperature, ultraviolet rays, etc.). It behooves clinicians to implement the best preservation methods for any type of elastic batches to ensure their optimal qualities overtime.

At least one article found that the forces of the elastics and their degradation did not match the manufacturing specifications, and that in vivo force loss was higher than in vitro samples, in both dry and with artificial saliva conditions [14]. Moreover, the reported specifications in terms of tension and degradation of elastics is different among manufacturers [10], so clinical oversight and expertise is important.

Latex elastics express their maximum force within the first hour, so maybe they are more suitable for less compliant patients as they have at least some traction [14]. Nonlatex elastics offer a more constant pull within eight hours, which could be an important aspect in patients who have bone issues that require a gentler and constant pull. The size of the elastic impacts the force decay, with longer elastics’ force diminishing faster over time than that of shorter elastics [17].

### 4.2. Study Limitations

The articles that presented a comparison between different brands of elastics of a single type (latex or nonlatex) were excluded from the pooled data as the verification methods were not uniform between the studies, while they were repeatable and homogeneous within the studies themselves [14,20,21].

There is a general lack of consistency and comparability in studies on this subject, with many articles that could not be considered due to their lack of rigorous scientific methodologies. Given the scarcity of comparative studies with uniform methods, it is advisable in future studies to expand the sample with dry condition studies, and possibly in artificial saliva, including in vivo vs. in vitro comparisons [21,22,23].

Considering that this review focused on the physical and chemical characteristics of the elastic bands and especially their force, as analyzed in vitro, it would be interesting to review and compare studies in which different elastic forces are analyzed as they affect the bone and periosteum, to evaluate the optimal force to apply to the teeth [23,24,25].

Given the scarcity of comparative studies with uniform methods, it was not possible to carry out a meta-analysis since only 2 of the 10 selected articles were suitable for comparison due to the type of sample and uniformity of the data [13,17].

## 5. Conclusions

This systematic review indicated that intraoral orthodontic elastics (IOEs) in general have the greatest loss of force during the first three hours. However, latex bands exerted the greatest force within the first hour. Regardless of size and manufacturers, elastic bands continue to have a decay of force at slower speed, at the interval marks of 6–8–12–24 h. According to the findings of force loss, the clinician may need to choose between an initial force much higher than desired vs. a force near the desired amount that will quickly decay to below the level required for the desired effect. Nonlatex rubber bands (elastomeric-based) need to be changed more frequently and regularly during the 24 h cycle. Another important aspect that surfaced from this systematic review is that, knowing that all IOEs undergo some levels of degradation, and in vivo results seems to differ from the manufacturers’ specification, it would be important for the manufacturers to print on the elastic bags a “best by” date, and the clinicians need to be more aware of best practices to preserve (warehouse) batches of IOE.

## Figures and Tables

**Figure 1 jpm-13-01495-f001:**
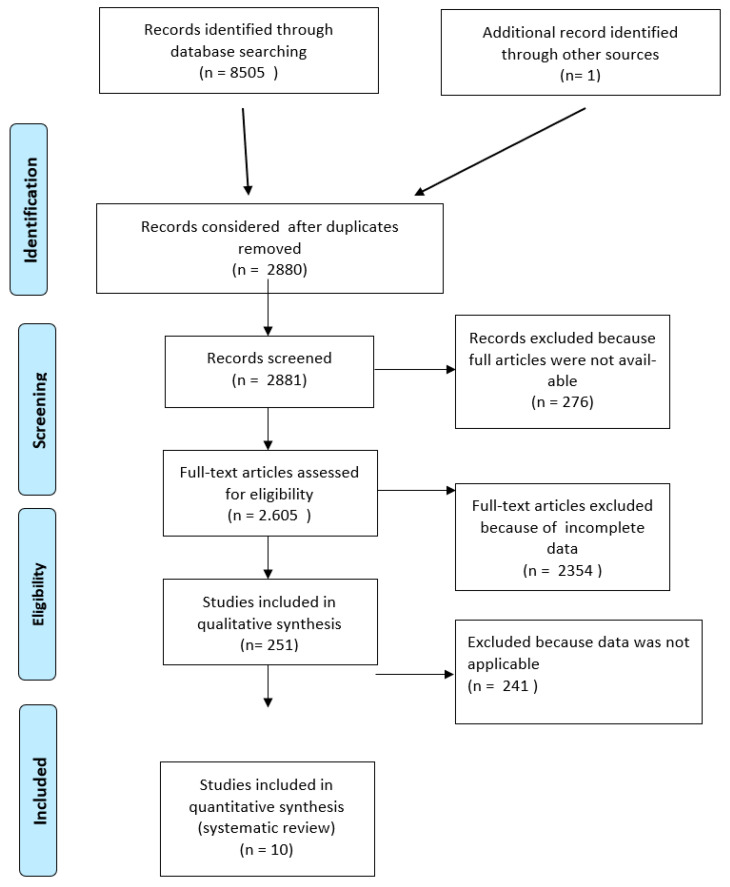
Flow chart: summary of the selection process according to the PRISMA guidelines.

**Table 1 jpm-13-01495-t001:** Summary of the 10 articles considered in the systematic review.

Title	Year	Authors	Size Elastic	Type Elastic
Evaluation of force degradation characteristics of orthodontic latex elastics in vitro and in vivo.	2007	Wang, T., Zhou, G., Tan, X., Dong, Y. [10].	3/16 in	Latex
Force degradation of orthodontic latex elastics analyzed in vivo and in vitro.	2019	Yang, L., Lu, C., Yan, F., Feng, J. [11].	1/4 in	Latex
Calibration of force extension and force degradation characteristics of orthodontic latex elastics.	2000	Kanchana, P., Godfrey, K. [12].	3/16, 1/4, and 5/16 in	Latex
Force extension relaxation of medium force orthodontic latex elastics.	2011	Fernandes, D.J., Fernandes, G.M., Artese, F., Elias, C.N., Mendes, A.M. [13].	3/16, 1/4, and 5/16 in	Latex
A comparison of orthodontic elastic forces: Focus on reduced inventory.	2017	Mansour, A.Y. [4].	1/4 and 3/16 in	Latex
Mechanical and biological comparison of latex and silicone rubber bands.	2003	Hwang, C.J., Cha, J.Y. [14].	Not specified	Latex and nonlatex
In vitro study of force decay of latex and non-latex orthodontic elastics.	2012	López, N., Vicente, A., Bravo, L.A., Calvo Guirado, J.L., Canteras, M. [15].	0.25 in and 4 oz	Latex and nonlatex
An in-vitro comparison of force loss of orthodontic non-latex elastics.	2014	Alavi, S., Tabatabaie, A.R., Hajizadeh, F., Ardekani, A.H. [6].	3/16 (medium)	Nonlatex
A study of force extension and force degradation of orthodontic latex elastics: An in vitro study.	2013	Gangurde, P.V., Hazarey, P.V., Vadgaonkar, V.D. [5].	Not specified	Latex
Orthodontic latex elastics: a force relaxation study.	2006	Gioka, C., Zinelis, S., Eliades, T., Eliades, G. [7].	3/16, 1/4, 5/16, and 3/8 in	Latex

**Table 2 jpm-13-01495-t002:** Synopsis of the articles considered for the systematic review.

Title	Year	Authors	Synopsis
Evaluation of force degradation characteristics of orthodontic latex elastics in vitro and in vivo.	2007	Wang, T., Zhou, G., Tan, X., Dong, Y. [10].	The most significant force degradation occurred in the first half hour, during both in vivo and in vitro studies, but the magnitudes of force loss were different.
Force degradation of orthodontic latex elastics analyzed in vivo and in vitro.	2019	Yang, L., Lu, C., Yan, F., Feng, J. [11].	The force degradation of latex elastic in vivo is much greater than that in both air and artificial saliva.
Calibration of force extension and force degradation characteristics of orthodontic latex elastics.	2000	Kanchana, P., Godfrey, K. [12].	There were significant differences in force extension and force degradation characteristics between different extensions and force magnitudes for the elastics of the different manufacturers.
Force extension relaxation of medium force orthodontic latex elastics.	2011	Fernandes, D.J., Fernandes, G.M., Artese, F., Elias, C.N., Mendes, A.M. [13].	The force decay pattern showed a notable drop-off of forces during 0 to 3 h, a slight increase in force values from 3 to 6 h, and a progressive force reduction over 6 to 24 h.
A comparison of orthodontic elastic forces: Focus on reduced inventory.	2017	Mansour, A.Y. [4].	The use of 1/4” diameter elastics is sufficient to cover the range of forces in orthodontic treatment.
Mechanical and biological comparison of latex and silicone rubber bands.	2003	Hwang, C.J., Cha, J.Y. [14].	The latex bands all followed a similar pattern of force degradation, whereas the silicone bands showed a greater increase in force decay as the extension length increased.
In vitro study of force decay of latex and nonlatex orthodontic elastics.	2012	López, N., Vicente, A., Bravo, L.A., Calvo Guirado, J.L., Canteras, M. [15].	Lancer^®^ nonlatex was the only type of elastic that did not show a significant decrease in its initial elastic characteristics at eight hours.
An in-vitro comparison of force loss of orthodontic non-latex elastics.	2014	Alavi, S., Tabatabaie, A.R., Hajizadeh, F., Ardekani, A.H. [6].	According to the initial force and force loss percentage over time, it is suggested to replace the nonlatex elastics several times a day.
A study of force extension an force degradation of orthodontic latex elastics: An in vitro study.	2013	Gangurde, P.V., Hazarey, P.V., Vadgaonkar, V.D. [5].	The degradation of strength in all types of rubber bands was equal to 20% on the first day and 5–10% on the second day.
Orthodontic latex elastics: a force relaxation study.	2006	Gioka, C., Zinelis, S., Eliades, T., Eliades, G. [7].	Latex elastics show force relaxation in the order of 25%, which consists of an initial high slope component and a latent part of decreased rate.

**Table 3 jpm-13-01495-t003:** The summary table of search strategy.

PICO Component	Details
Population	NA (they are all in vitro studies).
Interventions	Analysis of intraoral elastics in vitro.
Comparison	Intraoral elastics of different strengths, diameters, and materials.
Outcomes	The primary outcome was to evaluate how intraoral elastics behave in terms of tension strength and duration. The secondary outcome was to investigate the force degradation during the first hours of wear. The tertiary outcome was to assess how forces decayed.
Study design	The systematic review took into consideration original articles, comparative studies, and research articles, and once the final articles were selected, based on inclusion and exclusion criteria, the characteristics of the elastic bands were analyzed and expressed in terms of force, diameter, and material: Force express (oz); Force express (g); Diameter (in); Diameter (mm); Material.

**Table 4 jpm-13-01495-t004:** Summary of the risk of bias for each article included in the systematic review. Adapted from [15].

Risk of Bias	Incomplete Outcome Data	Selective Reporting	Other Bias
Wang 2007	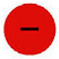	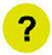	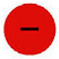
Yang, L. 2019	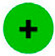	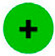	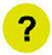
Kanchana 2000	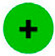	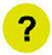	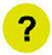
Fernandes 2011	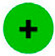	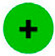	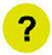
Mansour 2017	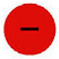	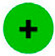	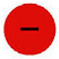
Hwang 2003	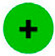	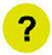	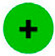
López 2012	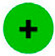	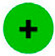	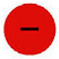
Alavi 2014	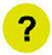	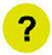	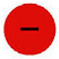
Gangurde 2013	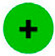	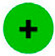	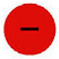
Gioka 2006	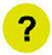	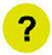	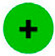

Key: 
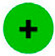
 —Low risk of bias; 
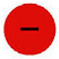
—High risk of bias; 
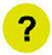
—Unclear risk of bias.

**Table 5 jpm-13-01495-t005:** Latex and nonlatex intraoral elastics mentioned in the selected articles.

Force Express [oz]	Force Express [g]	Diameter [in]	Diameter [mm]	Material
Light 2 oz	Light 57 g	1/8″	3.2 mm	Latex
Medium 4 oz	Medium 114 g	3/16″	4.7 mm	Nonlatex
Heavy 6 oz	Heavy 170 g	1/4″	6.4 mm	Silicon
X-Heavy 8 oz	X-Heavy 227 g	5/16″	7.9 mm	Not specified
Not specified	Not specified	3/8″	9.5 mm	Not specified

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
