# Peer review of "Evaluation of the Loss of Strength, Resistance, and Elasticity in the Different Types of Intraoral Orthodontic Elastics (IOE): A Systematic Review of the Literature of In Vitro Studies"

_jpm, 2023, doi:10.3390/jpm13101495_

Round 1

Reviewer 1 Report

Interesting article, good work, good preparation. However, as a reviewer, I have a few observations:

Abstract

Better to no use the same word twice in the same sentence. Elastic- instead use elastomeric elastic use  elastomeric based or  from elastomeric materials. And add to the first sentence elastic bands to be more clear.

Introduction

Line 47

Tweed ….[3]- but in references you have

Quinzi V, Scibetta ET, Marchetti E, et al. Analyze my face. J Biol Regul Homeost Agents. 2018;32(2 379

Suppl. 1):149-158.

Line 66

It would be worth adding that these Med Devs must meet the standard, it states what strength they must have after 24 hours and similar information

ISO 21606:2022(en) Dentistry — Elastomeric auxiliaries for use in orthodontics.

This is a cosmetic note, but in Table 2 and the Table, add reference numbers next to the authors' names

This is also a detail, but Table 1 should be before Table 2 and Table 3.

Line 178

Force express (gr)- if you are using grams it should be [g] , rest of the unites  in [.]

 Results

Your work has a title …..Evaluation of the loss of strength, resistance and elasticity in

the different types of intraoral elastics: a systematic review of the literature on in vitro studies

but  you look at articles  in vivo studies, So maybe it will be good to change the title and  short it :

Evaluation of the loss of strength, resistance and elasticity in  the different types of intraoral elastics: a systematic review of the literature ?

Discussion

You use the words alternately to start the sentence according, or regarding. You can also use others, e.g. other authors...... also reached the same conclusions.... Results described in the work..... So as to use the wealth of words of the English language.

The discussion could also include information about the mechanisms of degradation of materials in the oral cavity, why it happens, and what happens to latex or latex-free materials or silicones.

Reference

2. Inchingolo AD, Patano A, Coloccia G, Ceci S, Inchingolo AM, Marinelli G, Malcangi G, Di Pede C, Garibaldi M, Ciocia AM, Mancini A, Palmieri G, Rapone B, Piras F, Cardarelli F, Nucci L, Bordea IR, Scarano A, Lorusso F, Giovanniello D, Costa S, Tartaglia GM, Di Venere D, Dipalma G, Inchingolo F. Treatment of Class III Malocclusion and Anterior Crossbite with Aligners: A Case Report. Medicina (Kaunas). 2022 Apr 27;58(5):603. doi: 10.3390/medicina58050603. PMID: 35630020; PMCID: 377PMC9147027. 378

3. Quinzi V, Scibetta ET, Marchetti E, et al. Analyze my face. J Biol Regul Homeost Agents. 2018;32(2)Suppl. 1):149-158.

good luck in further research!

small improvement need

Author Response

Please find it in the attachment

Reviewer 2 Report

 Evaluation of the loss of strength, resistance and elasticity in the different types of intraoral elastics: a systematic review of the literature on in vitro studies

The introduction is comprehensive and accurate, however, attention to punctuation should be focused: ex page 2 line 71, “.,”.

The aim of this study was to evaluate, through a systematic review of the literature 85 of in vitro studies, the properties of intraoral elastics, the loss of strength, resistance, and elasticity, however, the connection to clinical utility should be described. 

Material and method: The PRISMA guidelines were followed. 

The protocol was registered in PROSPERO.

The search strategy from the Pubmed database should be presented as a table. 

The Flow Chart 1 should provide also the reason for study exclusion.

A table with the assessment of Risk of Bias should be presented. 

Results and discussion: in what way do the forces generated by the rubber bands not consistently correspond to the loads specified by the manufacturers? Bring arguments. 

Discussion: page 8 line 241 – is a repetition of the results. Please add other studies in the discussion section. 

When discussing Lancer elastics, please do consider also other manufacturers, otherwise do not include this statement. 

In the discussion section, the year is not necessary to be put into ().

When discussing elastic properties, the elastics should be grouped according to their dimensions (ex ¼, 3/16, aso). 

Considering that this review focused on the physical and chemical characteristics of the elastic bands in vitro, in what manner does the saliva influence the outcome? 

References cannot be shown in the conclusion section. 

Manufacturers should not be listed in the conclusion section. 

The conclusion should be more clearly defined. 

Author Response

Please find it in the attachment

Round 2

Reviewer 2 Report

Evaluation of the loss of strength, resistance and elasticity in the different types of intraoral elastics: a systematic review of the literature on in vitro studies – revision 1  

The aim of this study was to evaluate, through a systematic review of the literature of in vitro studies, the properties of intraoral elastics, the loss of strength, resistance, and elasticity, however, the connection to clinical utility has not been described

Material and method: The PRISMA guidelines were followed. 

The protocol was registered in PROSPERO.

The search strategy from the Pubmed database has not been presented as a table. It is required to see the search strategy. 

A table with the assessment of Risk of Bias(ROB)should be presented, not table 2, which refers to another topicA table with the ROB according to Cochrane must be added (low, medium or high risk).

Author Response

To the Editor in Chief of  Journal of Personalized Medicine

Dear Editor,

We  sending the manuscript revised entitled:  Evaluation of the loss of strength, resistance and elasticity in the different types of intraoral elastics: a systematic review of the literature on in vitro studies

Saccomanno S.1 Quinzi V.1 Coceani Paskay L.  2  Caccone L.1 Rasicci L.1Iommazzo B.1 Di Giandomenico D.1 Marzo G.1

1)Department of Health, Life and Environmental Science, University of L’Aquila, Piazza Salvatore Tommasi, 67100 L’Aquila, Italy

3)Academy of Orofacial Myofunctional Therapy (AOMT), Pacific Palisades, USA

Sabina Saccomanno

Email: sabinasaccomanno@hotmail.it

Phone +39 339 4153290

We thank the reviewers for their generous comments on the manuscript.

Neither the manuscript nor any parts of its content are currently under consideration or published in another journal.

We believe that the manuscript is now suitable for publication in your journal.

Dr. Sabina Saccomanno

On behalf of all the authors.

We have put the corrections in blue

We thank the reviewers for their generous comments on the manuscript.

Neither the manuscript nor any parts of its content are currently under consideration or published in another journal.

We believe that the manuscript is now suitable for publication in your journal. Dr. Sabina Saccomanno
On behalf of all the authors.

Evaluation of the loss of strength, resistance and elasticity in the different types of intraoral elastics: a systematic review of the literature on in vitro studies – revision 1  

The aim of this study was to evaluate, through a systematic review of the literature of in vitro studies, the properties of intraoral elastics, the loss of strength, resistance, and elasticity, however, the connection to clinical utility has not been described

Done :  added before of study limitation

Summary of connections to clinical utility

The studies taken into consideration for this systematic review highlight some clinically useful information:
Degradation over time of all types of elastics seems inevitable (Lopez et al). It would be ideal if the manufacturers would provide a “best by” date or preservation recommendations for elastics (such as light exposure, temperature, ultraviolet rays etc). It behooves clinicians to ensure the best preservation methods for any type of elastic batches to ensure their optimal qualities overtime.
At least one article found that the forces of the elastics and their degradation does not match the manufacturing specifications, and that in vivo force loss is higher than in vitro samples, both in dry or with artificial saliva conditions, (Yang et al). Moreover, the reported specifications in terms of tension and degradation of elastics is different among manufacturers (Kanchana and Godfrey), so clinical oversight and expertise is important.
Latex elastics express their maximum force within the first hour, so maybe they are more suitable for less-compliant patients as they get at least some traction (Yang, Lu, Yan, Feng).
Non-latex elastics offer a more constant pull within eight hours, which could be an important aspect in patients who have bone issues that require a gentler and constant pull.
The size of the elastic impacts the force decay, with longer elastics’ force diminishing faster over time than that of shorter elastics (Hwang & Cha).

And in

Conclusions

. Another important aspect that surfaced from this systematic review is that, knowing that all IOE undergo some levels of degradation, and in vivo results seems to differ from the manufacturers’ specification, it would be important for the manufacturers to print on the elastic bags a “best by” date and the clinicians need to be more aware of best practices to preserve (warehouse) batches of IOE. 

Material and method: The PRISMA guidelines were followed. 

The protocol was registered in PROSPERO.

The search strategy from the Pubmed database has not been presented as a table. It is required to see the search strategy. 

Done

Table The summary  table of   search strategy

PICO

COMPONENT

DETAILS

Population 

NA (they are all in vitro studies)

Interventions 

Analysis of intraoral elastics in vitro

Comparison 

Intraoral elastics of different strengths, diameters and materials.

Outcomes

The primary outcome was to evaluate how intraoral elastics behave in terms of tension strength and duration.

The secondary outcome was to investigate the force degradation during the first hours of wearing.

The tertiary outcome was to assess how forces decayed.

Study design

The systematic review took into consideration original articles, comparative studies, research  articles and once the final articles were selected, basedon inclusion and exclusion criteria, the characteristics of the elastic bands were analyzed and expressed in Force, Diameter and Material. 

Force express [oz]

Force express [g]

Diameter [in]

Diameter [mm]

Material

A table with the assessment of Risk of Bias(ROB)should be presented, not table 2, which refers to another topic. A table with the ROB according to Cochrane must be added (low, medium or high risk).

DONE

Risk of biais

Incomplete outcome data

Selective  Reporting

Other bias

Wang 2007

Yang L, 2019

Kanchana 2000

Fernandes 2011

Mansour2017

       Hwang 2003

López 2012

Alavi 2014

Gangurde 2013

Gioka 2006

KEY

LOW RISK OF BIAS

HIGH RISK OF BIAS

UNCLER  RISK OF BIAS